# Attention-Based Deep Feature Fusion for the Scene Classification of High-Resolution Remote Sensing Images

**Ruixi Zhu** , **Li Yan \*, Nan Mo and Yi Liu**

School of Geodesy and Geomatics, Wuhan University, 129 Luoyu Road, Wuhan 430079, China

\* Correspondence: lyan@sgg.whu.edu.cn

**Abstract:** Scene classification of high-resolution remote sensing images (HRRSI) is one of the most important means of land-cover classification. Deep learning techniques, especially the convolutional neural network (CNN) have been widely applied to the scene classification of HRRSI due to the advancement of graphic processing units (GPU). However, they tend to extract features from the whole images rather than discriminative regions. The visual attention mechanism can force the CNN to focus on discriminative regions, but it may suffer from the influence of intra-class diversity and repeated texture. Motivated by these problems, we propose an attention-based deep feature fusion (ADFF) framework that constitutes three parts, namely attention maps generated by Gradient-weighted Class Activation Mapping (Grad-CAM), a multiplicative fusion of deep features and the center-based cross-entropy loss function. First of all, we propose to make attention maps generated by Grad-CAM as an explicit input in order to force the network to concentrate on discriminative regions. Then, deep features derived from original images and attention maps are proposed to be fused by multiplicative fusion in order to consider both improved abilities to distinguish scenes of repeated texture and the salient regions. Finally, the center-based cross-entropy loss function that utilizes both the cross-entropy loss and center loss function is proposed to backpropagate fused features so as to reduce the effect of intra-class diversity on feature representations. The proposed ADFF architecture is tested on three benchmark datasets to show its performance in scene classification. The experiments confirm that the proposed method outperforms most competitive scene classification methods with an average overall accuracy of 94% under different training ratios.

**Keywords:** remote sensing; scene classification; attention maps; multiplicative fusion of deep feature; center loss

## 1. Introduction

It is increasingly significant to use high-resolution remote sensing images (HRRSI) in geospatial object detection [1,2] or land-cover classification tasks due to the advance of remote sensing instruments. As we all know, scene classification (that classifies scene images into diverse categories according to the semantic information they contain), has been widely applied to land-cover or land-use classification of HRRSI [3–6]. Nevertheless, it is difficult to classify the scene images effectively due to various land-cover objects and high intra-class diversity [7,8]. Therefore, features that are used to describe scene images are important for scene classification of HRRSI.

Features that are used to describe the scene images are mainly divided into three types by [9]. They include handcrafted features, unsupervised learning-based features and deep features. The literature of feature representations for scene classification is described in Section 2.1. Although the deep feature-based methods have achieved great success in scene classification, they assume that each object equally contributes to the feature representations of a scene image [10–12].

However, scene images of HRRSI contain more diverse types of objects compared with those in nature images [13] and not all objects in scene images are useful for recognizing the scenes [14]. As shown in Figure 1a, the cars and roads are important while the trees and houses are unimportant for classifying the freeway scenes. In Figure 1b,c, recognizing the airplane in the airport scene and tennis court in the tennis court scene can assist the scene classification, since the airplanes and tennis courts are the indispensable parts of airport and tennis court scenes. As a result, more emphasis should be laid on those important objects, and less emphasis should be laid on redundant objects when representing scene images. For this reason, the visual attention mechanism is studied in the CNN over recent years [15–21], and the literature on the attention mechanism is shown in Section 2.2. In the attention mechanism, some salient regions selected from the entire image rather than the entire image are processed by the visual attention mechanism at once. Computation cost can be reduced, and classification results can be improved by focusing on the important regions in scene images [18]. Despite the impressive results achieved by the CNN architectures that are incorporated into the attention mechanism, they still pose various challenges.

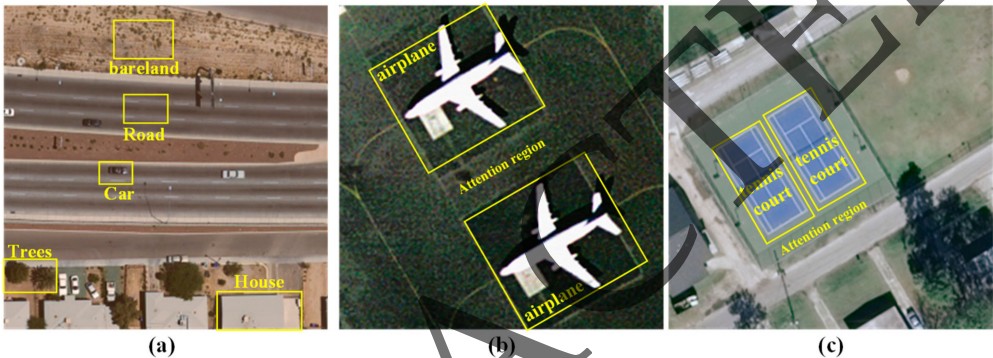

**Figure 1.** The salient and redundant objects in scenes and the attention mechanism. (**a**) Useful objects, including roads and cars and other redundant objects for classifying the freeway category. (**b**) Attention regions of scenes for airports. (**c**) Attention regions of scenes for tennis courts.

First of all, they still suffer from high intra-class variations existing in scene images of HRRSI [22]. That is because diverse seasons, locations or sensors may lead to highly different spectral characteristics of scene images with the same category [23], as shown in Figure 2. Therefore, it is hard to detect salient regions well due to highly different spectral characteristics of scene images belonging to the same category.

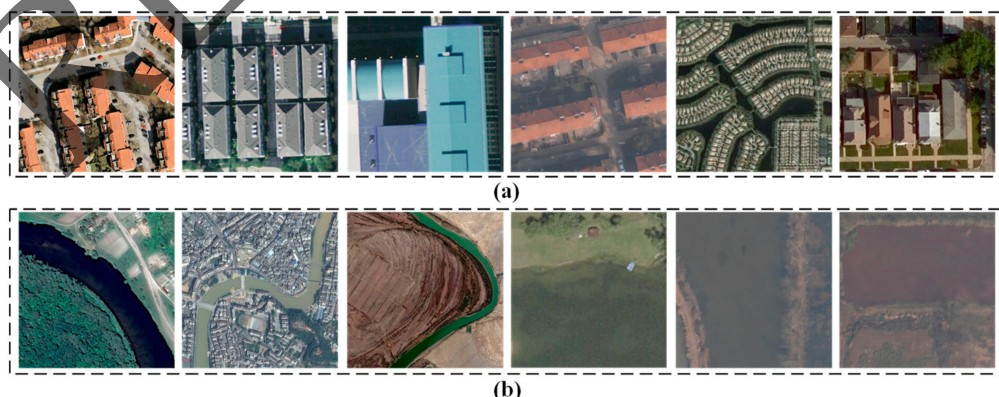

**Figure 2.** The intra-class diversity of remote sensing scene images. (**a**) Residential scene images. (**b**) River scene images.

Secondly, the existing attention mechanism methods assume that the salient regions can represent the label information of the scene images well [24]. But they ignore that the scene images of repeated

texture do not satisfy the assumption. As shown in Figure 3, the key regions derived from the attention mechanism are usually located in the center of images for scenes of repeated texture. But all objects in the entire images are equally important for classifying scene images, since there exists no redundant objects in these scenes.

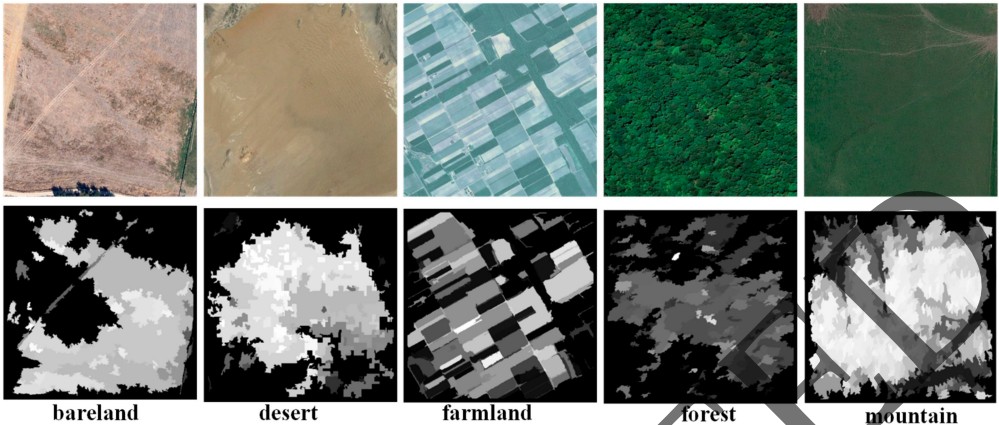

**Figure 3.** Attention maps of scenes of repeated texture.

In this paper, an attention-based deep feature fusion (ADFF) method for the scene classification of HRRSI is proposed to handle two challenges mentioned above. Firstly, the attention maps are generated from Grad-CAM algorithm to provide key regions for feature representations, and a CNN model is trained for RGB images. Then features derived from CNN model and attention maps are fused to take scenes of repeated texture and key regions into consideration. Finally, the center loss function is combined with the cross-entropy loss to reduce the influence of intra-class diversity on representing scene images.

The four major contributions are depicted as follows:

1. We propose to make attention maps related to original RGB images an explicit input component of the end-to-end training, aiming to force the network to concentrate on the most salient regions that can increase the accuracy of scene classification.
2. We design multiplicative fusion of deep features by combining features derived from the attention map with those from the original pixel spaces to improve the performance of these scenes of repeated texture.
3. We propose a center-based cross-entropy loss function to better distinguish scene images that are easily confused and decrease the effect of intra-class diversity on representing scene images.
4. The proposed ADFF framework is evaluated on three benchmark datasets and achieves state-of-the-art performance in the case of limited training data. Therefore, it can be applied to the land-cover classification of large areas when training data is limited.

The rest of this paper is organized, as follows. Section 2 summarizes the literature of feature representations, the attention mechanism and feature fusion. The details of the proposed ADFF algorithm are depicted in Section 3. Section 4 introduces the dataset description, experimental setup and results. The experimental results are analyzed in Section 5. The conclusions with a potential direction are presented in Section 6.

## 2. Related Work

### 2.1. Feature Representation

The previous scene classification methods are mainly divided into three types, according to the used feature representations. The three categories of features include handcrafted features, unsupervised learning-based features and deep features [9].

Handcrafted features are aimed to design features for scene classification manually. They represent the scenes images with their primary characteristics, including shape, texture, spectral information, or a fusion of these information. Histogram of oriented gradients [25–27], texture features [28,29] and spectral features are the most widely used among all these features. Nevertheless, the inadequate description of semantic information hidden in scenes may constrain the performance of these features.

The drawbacks of handcrafted features can be overcome by unsupervised learning-based features that have been studied by many researchers [30–32]. The input of unsupervised learning-based features is some handcrafted features, while the statistics of handcrafted features are output. This family of features covers probabilistic latent semantic analysis (PLSA), latent dirichlet allocation (LDA), bag of visual words (BOVW) and principal component analysis (PCA). Unsupervised learning-based features can obtain more discriminative features compared with handcrafted features, which may lead to better classification performance. However, the ability to distinguish similar scene images cannot be guaranteed by most unsupervised learning-based features because the hidden semantic information is not made full use of.

Deep features have attracted more and more tremendous attention because of the development of GPU and an increasing number of remote sensing training samples [33,34]. These methods have two important advantages. For one thing, the semantic features can be learned from original images automatically via CNN [35–38]. Therefore, we only need to modify the structure of CNN architectures rather than manually design the features, which can save the human resources. For another, deep features are more discriminative in distinguishing similar scene categories compared with the other two types of features [39].

### 2.2. Attention Mechanism

The predecessor of the attention mechanism is the saliency detection that calculates a saliency value for each pixel in the scene image. The saliency detection method that is aimed at assigning each pixel in the scene images with a saliency value assumes that salient regions usually correspond to the true scene labels. Several studies have been done in the field of saliency detection. Zhang et al. [24] use all detected salient regions and some unimportant regions to represent each scene image and eliminate some unimportant objects. Zhang et al. [40] propose to extract crucial objects by the saliency maps, which can increase the accuracy of land-cover classification.

Although the attention mechanism is similar to saliency detection superficially, a difference still exists between the saliency detection and attention mechanism. The saliency detection methods cannot distinguish the importance of each object, since they obtain the salient regions of scene images by the texture information. The attention mechanism methods learn important regions or parts from the entire images by adjusting the supervision signal continuously during the training process. The largest discrepancy between the saliency detection and attention mechanism is that the attention mechanism concentrates on learning while saliency detection pays attention to calculation. In the case of sufficient training data, learning can deliver better performance for the attention mechanism compared with saliency detection methods.

In the context of the attention mechanism for image classification, three common techniques are used to compute attention maps given category labels, namely CAM [41], Grad CAM [42], and Grad-CAM++ [43]. The drawbacks of the CAM approach are that it is inflexible and requires architecture modification for generating trainable attention. Although Grad CAM++ can produce a better class activation map by modifying the way weights are computed, its high computational cost in calculating the second and third derivatives makes it impractical. Grad CAM method does not need architecture modification or re-training, and it only requires calculating the first derivatives, which may save much computation cost. Therefore, Grad CAM is used to generate attention maps in this paper.

## 2.3. Feature Fusion

Feature fusion is a significant means that can comprehensively achieve complementary advantages of different features and obtain robust and accurate recognition results. Feature fusion is often used in deep learning models. Feichtenhofer et al. [44] propose to fuse deep features after the convolutional layer in both the softmax and RELU layers by a spatial and temporal feature fusion method. Chaib et al. [45] propose to fuse deep features by discriminant correlation analysis. Zhao et al. [46] fuse the spectral information with structural information hidden in scene images to increase the discriminative ability of features. Cheng et al. [26] propose to obtain the BOVW feature vector by combining the local features with global features. Chowdhury et al. [47] propose to calculate the outer product of deep features extracted from different convolutional layers in order to fuse deep features in the bilinear CNN. Jiang et al. [48] propose to fuse the spatial and motion features for classification to achieve deep feature fusion. Features are fused in different CNNs in the deep feature fusion network proposed by Bodla et al. [49]. The feature fusion methods can complement each other by fusing features. They can reduce the influence of scenes of repeated texture on the feature representations in this paper.

## 3. Materials and Methods

### 3.1. Overall Architecture

As shown in Figure 4, the proposed ADFF approach consists of three novel components, namely the network that generates attention maps by the Grad-CAM, a multiplicative fusion of deep features and the center-based cross-entropy loss function. Sections 3.2–3.4 elaborate each novel component of the ADFF framework.

The left part of our framework shows the structure we generate the attention map. We fine-tune the pre-trained ResNet-18 model [50] on existing samples because the features learnt from fine-tuned architectures are more suitable for classifying HRRSI. Then we generate attention map for all images by the Grad-CAM algorithm in Section 3.2. The right half of our framework in Figure 4 shows our end-to-end learning model, including multiplicative fusion of features extracted from CNN models and spatial feature transformer (SFT) in Section 3.3 and integration of cross-entropy loss and center loss functions in Section 3.4. Algorithm 1 summarizes the process of the ADFF framework.

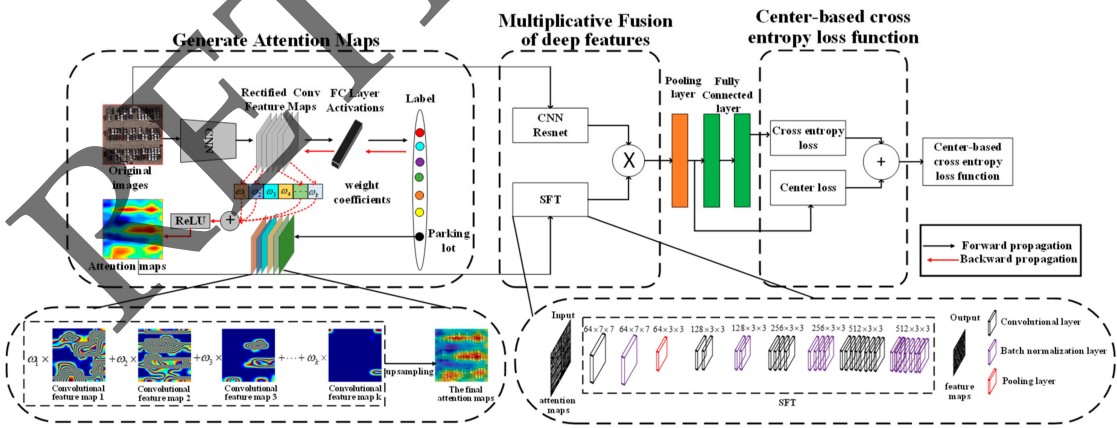

**Figure 4.** The overall architecture of the proposed method. The different color blocks represent different network structure layers, respectively.

---

**Algorithm 1.** The procedure of ADFF

---

**1**     **Step 1 Generate attention maps**
**2**     **Input:** The original images *X* and their corresponding labels *Y*.
**3**     **Output:** Attention maps $X_{am}$.
**4**     Fine-tune ResNet-18 model on training datasets.
**5**     Forward inference full image *X*.
**6**     Calculate weight coefficients $\alpha$ from Equation (1).
**7**     Obtain gray scale saliency map $X_{sm}$ from Equation (2).
**8**     **Return** attention map $X_{am}$ by upsampling $X_{sm}$ to the size of *X*.
**9**     **Step 2 End-to-end learning**
**10**    **Input:** The original image *X* and the attention maps $X_{am}$
**11**    **Output:** Predict probability **P**
**12**    **While** E*poch=1, 2, . . . , N* do
**13**    Fuse features derived from CNN and SFT that are trained from *X* and $X_{am}$ respectively.
**14**    Predict probability of images $P = f(X, X_{am})$ by the fused features.
**15**    Calculate the total loss function $L_{total}$.from Equation (6)
**16**    Update parameter $\theta$ through back propagating the loss $L_{total}$ in 15
**17**    **End while**
**18**    **Return** Predict probability **P**

---

### 3.2. Attention Maps Generated by Grad-CAM Approach

For scene classification of HRRSI, some objects of scene images are redundant, which may negatively influence the representation of scene images. The HRRSI used in this paper are described in Section 4. Therefore, salient objects need to be detected from the scene images in order to reduce the influence of insignificant objects on representing the scene images. In this paper, attention maps that are originally used to explain the predictions of the CNN model are introduced to extract key regions. We resort to the Grad-CAM approach to produce attention maps on all training and test images.

The approach to generating attention maps contains two steps, including forward propagation and backward propagation. For forward propagation, we fine-tune the pre-trained ResNet networks on remote sensing training data. For backward propagation, we mainly use the Grad-CAM to generate the attention maps that can assist the scene classification of each particular dataset.

In Grad-CAM, we first compute the neuron importance weights of class *c* $\alpha_k^c$, as shown in Equation (1).

$$\alpha_k^c = \frac{\sum_i \sum_j \frac{\partial y^c}{\partial A_{i,j}^k}}{Z}, \tag{1}$$

where $y^c$ denotes the score of class *c* and $A^k$ denotes the *k*-th convolutional feature maps derived from the last convolutional layer. $\alpha_k^c$ represents the relative importance coefficient of *k*-th convolutional feature maps for *c*-th category and *Z* represents a feature map's pixel number.

Then convolutional feature maps are combined with different weights $\alpha_k^c$. Finally, we get the class-discriminative attention map $L_{GRAD-CAM}^c$, as shown in Equation (2) and Figure 5, by putting the combined feature maps into an RELU layer,

$$L_{GRAD-CAM}^c = \text{Re}LU(\sum_k \alpha_k^c A^k). \tag{2}$$

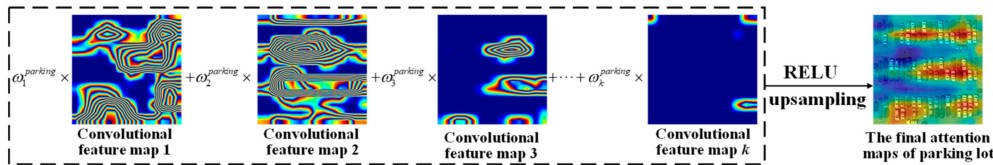

**Figure 5.** The convolutional feature maps and lass-discriminative attention map in the Grad-CAM method.

The attention maps provided by the Grad-CAM approach can offer information about salient regions that are important for representing scene images and reduce the negative influence of unimportant objects on feature representations.

### 3.3. Multiplicative Fusion of Deep Features Derived from CNN and SFT

Using only the original RGB images as the input of CNN architecture may suffer from redundant objects of scene images while only the attention maps as the input may cause a lower performance in scene images of repeated texture. Feature fusion is an efficient solution to this problem. Therefore, we propose a simple, but effective, multiplicative fusion of deep features from two different streams for the scene classification of HRRSI.

As can be seen in Figure 4, the first stream feeds original RGB images into the CNN architecture. The structure of CNN architecture that is trained in this stream is consistent with the ResNet-18 network structure.

The second stream utilizes the attention maps as input to the train the designed spatial feature transformer (SFT) network, since it is parameter-efficient in extracting valuable information from attention maps and the features output by the SFT are easily fused with those from the CNN because of the same feature dimension. The architecture of SFT is presented in Figure 6. SFT contains four convolutional layers, four batch normalization layers and one max-pooling layer only following the first batch normalization layer. The first convolutional layer has 64 filters of size $7 \times 7$ with a stride of two pixels and a padding of three pixels. The stride and padding of other convolutional layers are set as 2 and 1 pixel respectively. The second, third, and fourth convolutional layers have 128, 256 and 512 filters with a size of $3 \times 3$. The batch normalization layers are consistent with the kernel of the convolutional layer they are connected to. Max-pooling is carried out over a $3 \times 3$ window with a stride of 2.

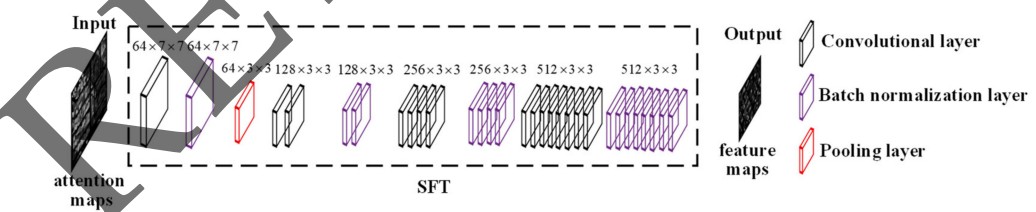

**Figure 6.** The spatial feature transformer.

When deep discriminative features are obtained from CNN and SFT respectively, we use multiplicative fusion functions shown in Equation (3) for high-dimensional deep feature fusion,

$$y_d^{mul} = X_{i,j,d}^{rgb} \times X_{i,j,d}^{grad-cam}. \tag{3}$$

In Equation (3), $i, j \in [7, 7]$ and $d$ is the feature dimension. The number of channels in $y$ is still 512.

The fused features consider both the salient objects and increased discriminative ability in scenes of repeated texture to make the fused features better differentiate scene images of repeated texture.

*3.4. The Center-Based Cross Entropy Loss Function*

Large intra-class differences caused by diverse natural environments, climates, sensors or latitudes may exist in scene images of HRRSI. Therefore, the cross-entropy loss is combined with the center loss to form the proposed center-based cross-entropy loss function so as to reduce the effect of within-class diversity.

Generally speaking, the cross-entropy loss is frequently applied to the scene classification of HRRSI, since it can evaluate the difference between the probability distribution of true labels and that of predicted labels [51,52], which may increase the discriminative ability of the CNN. Equation (4) shows the cross-entropy loss function.

$$L_s = -\sum_{i=1}^{m} \log \frac{e^{W_{y_i}^T x_i + b_{y_i}}}{\sum_{j=1}^{n} e^{W_j^T x_i + b_j}}, \tag{4}$$

where *n* is the category number and *m* is the number of training samples. $W_j \in \mathbb{R}^d$ represents weights of the last fully connected layer in the *j*-th column. $b \in \mathbb{R}^n$ represents the bias and $x_i \in \mathbb{R}^d$ represents the deep features derived from *i*-th image that is with the category $y_i$.

Although the cross-entropy loss function may increase the discriminative ability, it assumes that difficult samples and easy samples are of the same importance for training a CNN. Therefore, the cross-entropy loss function may deliver poor performance in classifying some difficult samples that are with high intra-class diversity and inter-class similarity. Center loss function [53] is introduced, as shown in Equation (5), in order to increase the discriminative ability of CNN in difficult samples while keeping the ability of features in distinguishing easy samples,

$$L_c = \frac{\sum_{i=1}^{m} \left\| x_i - c_{y_i} \right\|_2^2}{2}. \tag{5}$$

The $c_{y_i} \in \mathbb{R}^d$ represents the average value of all deep features in each mini-batch belonging to the category $y_i$. The mini-batch stochastic gradient descent (SGD) is used in the center loss to optimize the CNNs rather than optimize on the entire training dataset, which can reduce the computational cost. However, some scene images used for calculating the centers may not be predicted correctly.

Therefore, in order to learn a more discriminative CNN, we combine the cross-entropy loss with center loss. The proposed center-based cross-entropy loss can be given in Equation (6)

$$L_{total} = L_s + \lambda L_c, \tag{6}$$

where $\lambda$ is hyper-parameters that control a balance between the center loss and cross-entropy loss.

Features of different categories will be far away and those with the same label will be close if the weights of ADFF are backpropagated with the center-based cross-entropy loss. Then the influence of intra-class variations on feature representations will be reduced.

## 4. Experimental Results and Setup

In this section, descriptions of three datasets used to test the ADFF framework in this paper, the compared baseline approaches, the implementation details, along with evaluation metrics are given.

*4.1. Description of Datasets and Implementation Details*

4.1.1. Dataset Description

In order to evaluate the performance of the ADFF framework, three datasets, including UC Merced, Aerial Image Dataset (AID) and NWPU-NESISC45 dataset, are used. The details of three

datasets for evaluating the proposed ADFF framework are illustrated in Table 1. Then each dataset will be introduced separately.

**Table 1.** Summarization of three datasets used in this paper.

| Details | UC Merced | AID | NWPU-RESISC45 |
|---|---|---|---|
| Size of each patch | $256 \times 256$ | $600 \times 600$ | $256 \times 256$ |
| Spatial resolution | 0.3 m | 0.5–8 m | 0.2–30 m |
| The number of classes | 21 | 30 | 45 |
| Images per class | 100 | 200–400 | 750 |
| Total images | 2100 | 10000 | 31500 |

1.  **UC Merced Land-use Dataset** [54]: This dataset is the most classical scene classification benchmark dataset that is annotated from a publicly available aerial image. It contains 100 scene images of each category in 21 land-cover categories, with a spatial resolution of 0.3 m. These scenes are all with an image size of $256 \times 256$. Figure 7 shows the example images of 21 land-cover categories. This dataset can be obtained at http://weegee.vision.ucmerced.edu/datasets/landuse.html.

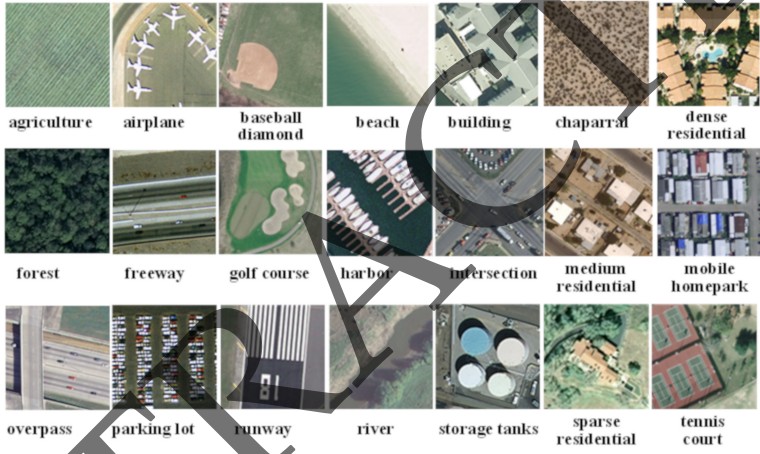

**Figure 7.** Example images of UC MERCED dataset.

2. **Aerial Image Dataset** [55]: This dataset is annotated by experts that are experienced in visual interpretation. It contains a total of 10,000 scene images of 30 land-cover categories. These scenes are all with a $600 \times 600$ image size. Ranging spatial resolutions and richer image variations can be offered in this dataset. The example images of each class in this dataset are shown in Figure 8. The big dataset is released at http://captain.whu.edu.cn/project/AID/.

3. **NWPU-RESISC45 dataset** [56]: This dataset is more complex than the other two datasets, since it contains 700 scene images for each category in 45 land-cover categories. These scenes are all with an image size of $256 \times 256$ and a spatial resolution of 0.3 m. The example images of all 45 categories are shown in Figure 9. The dataset can be obtained at the address of http://www.escience.cn/people/JunweiHan/NWPU-RESISC45.html.

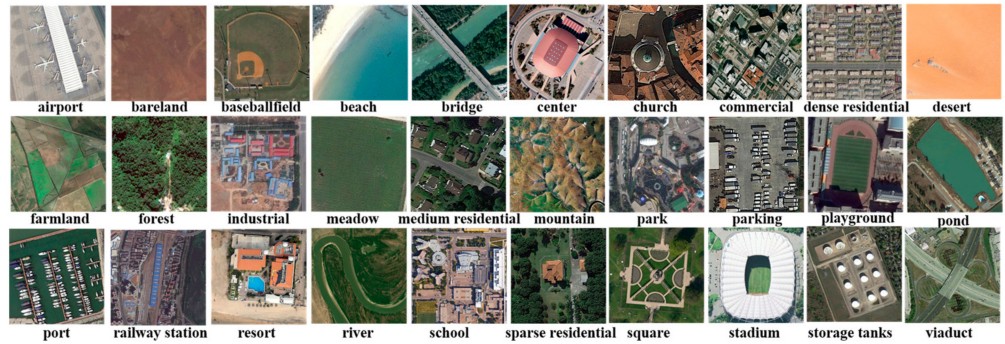

**Figure 8.** The example images of AID dataset.

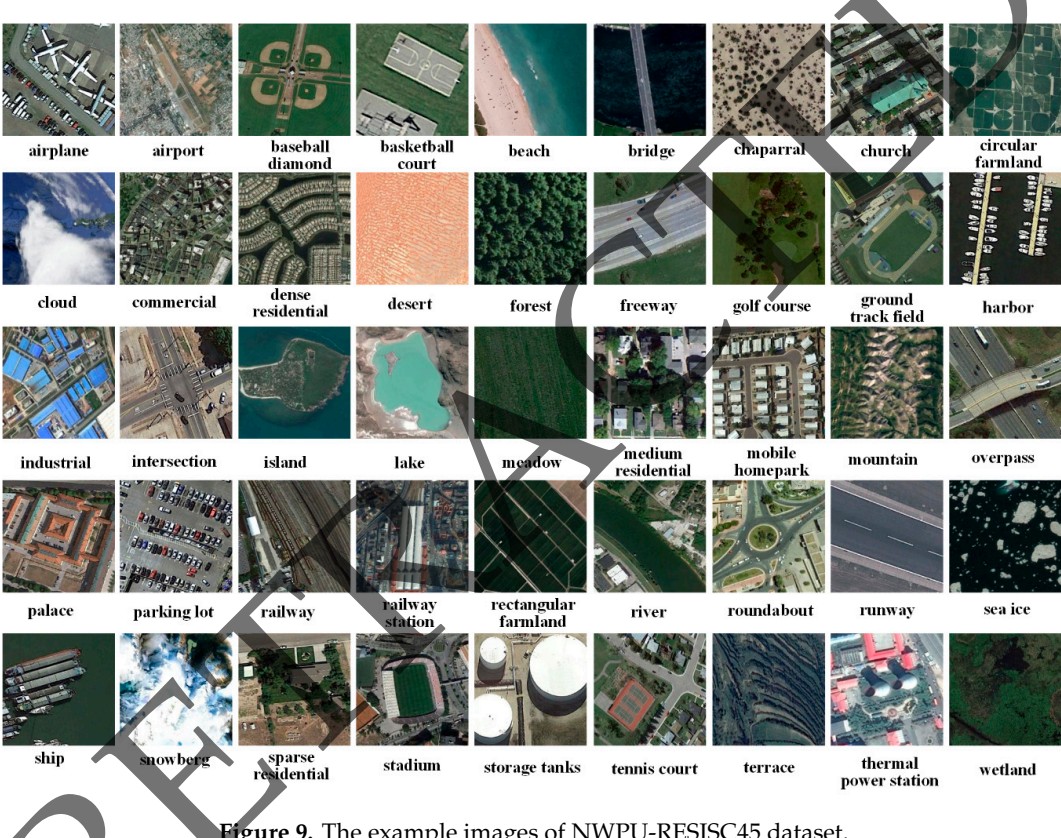

**Figure 9.** The example images of NWPU-RESISC45 dataset.

### 4.1.2. Compared Approaches and Implementation Details

The ADFF methods are compared with some baseline deep feature learning-based approaches and unsupervised feature learning approaches so as to demonstrate the effectiveness of the ADFF framework. Deep feature learning-based methods include CaffeNet with DCF [57], VGG-VD16 with DCF [57], TEX-Net-LF [58], CaffeNet [59], GoogleNet [60], VGG-16 [61], Multi-Branch Neural Network [62], triplet network [63], Hydra [64] and DenseNet [65]. Unsupervised feature learning methods BOCF [66], salM3LBP-CLM [10], scene capture [8] and color-boosted saliency-guided BOW [67] are also compared with the proposed ADFF approach to show the performance of the ADFF framework compared with the unsupervised feature learning methods. All scene classification results of the proposed ADFF framework have been released at https://drive.google.com/drive/folders/1-W8p7miChhrYP0ngsFqYwIhrsaGoy8Yq.

In this work, the pytorch framework is used to implement the proposed method. The hyperparameters used in the training stage are set by trial and error as follows. The batch size in the Adam optimizer is fixed as 128 to cater to the GPU memory. The initial learning rate is

fixed as 0.001 and decreased by 0.1 for every 10 epochs. The maximum epochs in the source domain, the parameter controlling the learning rate of centers α and balance between the cross-entropy loss and center loss λ are fixed to 40, 0.01 and 0.1 respectively by cross validation.

All CNN models are trained until the training loss function converged. At the same time, for a fair comparison, the same ratios of training samples are applied in the following experiments according to the experimental settings in works. The 50% and 80% training samples are experimented separately for evaluating the UC Merced dataset. For the AID dataset, 20% and 50% of images are chosen. In addition, 20% and 10% training ratio are used for the NWPU-RESISC45 dataset. Here, two training ratios are considered for each of the three datasets because the numbers of images in three datasets are highly different. A small ratio can usually satisfy the full training requirement of the models when a dataset has numerous scene images. In actual training, all training samples constitute 50% training dataset and 50% validation dataset. The remaining scene images except those in the training and validation dataset are considered as the test dataset. Five-fold cross validations are used in this paper to show the stability of the proposed ADFF framework, which means a random selection of the training samples from the whole dataset for five times.

Note that all images in the AID dataset are resized to 256 × 256 pixels from the original 600 × 600 pixels because of memory overflow in the training phase. All the implementations are evaluated on a windows 7 system with one 3.6 GHz 8-core i7-4790CPU and 16 GB memory. Moreover, a GPU of NVIDIA GTX 960 is used to increase the computing speed.

### 4.1.3. Evaluation Metrics

Three Evaluation metrics are used in this paper to evaluate the ADFF framework. They include confusion matrix, standard deviation and overall accuracy. Each evaluation metric is explained in detail.

**1. Overall Accuracy**: It is the ratio of accurately predicted scenes' quantity to all predicted scenes' quantity

**2. Standard deviation**: It is a metric that is used to quantify the amount of variation or dispersion for a set of overall accuracies.

**3. Confusion Matrix**: This evaluation metric is widely utilized in scene classification. Each column of this matrix is the predicted labels, while each row of the matrix is the true labels. The major confusion of one method can be easily recognized from the matrix.

### 4.2. Experimental Results on the UC MERCED Dataset

As shown in Table 2, some competitive scene classification approaches are compared with the ADFF framework to prove the effectiveness of the ADFF in increasing classification accuracy. According to the results of Table 2, the ADFF framework achieves the highest OA of 97.53% and 96.05% for 80% and 50% training ratio among all methods. This demonstrates that the ADFF can learn more discriminative feature representations of scene images by combining features derived from original images and attention maps. The VGG-VD16 with DCF, CaffeNet with DCF and TEX-Net-LF can provide comparable classification accuracy, since they can distinguish test scene images with high inter-class similarity and intra-class variations well. CaffeNet, GoogleNet, VGG-16, Multi-Branch Neural Network and triplet network provide a relatively lower accuracy, since these CNN architectures may fail to make features of scenes with the same land-cover category close and those with diverse land-cover categories separate during the training procedure.

The unsupervised feature learning-based approaches perform worse than the deep feature learning-based approaches. The BOCF offers the lowest classification accuracy, since the statistics of convolutional features may lose some context information. The salM3LBP-CLM performs better than the BOCF, since it fuses global salM3LBP and local BOCF features in order to overcome the drawbacks of the BOCF to some extent. The scene capture method and color-boosted saliency-guided BOW deliver

performances comparable to salM3LBP because they incorporate discriminative label information or salient region information into the feature representations, respectively.

**Table 2.** Standard deviations and overall accuracy (%) of the attention-based deep feature fusion (ADFF) framework and all baseline algorithms with 80% and 50% training ratio in the UC-Merced dataset.

| Methods | Published Year | 50% Training Ratio | 80% Training Ratio |
|---|---|---|---|
| The Proposed ADFF | 2019 | 96.05 ± 0.56 | 97.53 ± 0.63 |
| CaffeNet with DCF | 2018 | 95.26 ± 0.50 | 96.79 ± 0.66 |
| VGG-VD16 with DCF | 2018 | 95.42 ± 0.71 | 97.10 ± 0.85 |
| TEX-Net-LF | 2017 | 95.89 ± 0.37 | 96.62 ± 0.49 |
| CaffeNet | 2017 | 93.98 ± 0.67 | 95.02 ± 0.81 |
| GoogleNet | 2017 | 92.70 ± 0.60 | 94.31 ± 0.89 |
| VGG-16 | 2017 | 94.14 ± 0.69 | 95.21 ± 1.20 |
| Multi-Branch Neural Network | 2018 | 91.33 ± 0.65 | 93.56 ± 0.87 |
| Triplet Network | 2015 | 93.53 ± 0.49 | 95.70 ± 0.60 |
| BOCF | 2018 | 89.31± 0.92 | 92.50± 1.18 |
| salM3LBP-CLM | 2017 | 91.21 ± 0.75 | 93.75 ± 0.80 |
| Scene Capture | 2018 | 91.11 ± 0.77 | 93.44 ± 0.69 |
| Color-Boosted Saliency-Guided BOW | 2018 | 91.85 ± 0.65 | 93.71 ± 0.57 |

The confusion matrices of the ADFF method under the training ratio of 80% and 50% is depicted in Figure 10. As shown in Figure 10a, 18 categories achieve accuracies greater than 93%, and six categories achieve an accuracy of 100%. In addition, the classes of "medium residential", "storage tanks" and "tennis court" achieve a relatively lower accuracy of 93%. This may have resulted from the fact that the three classes are easy to be confused with classes that are with the same components building, but with different building structures. When the training ratio reduces from 80% to 50%, the accuracies of 17 categories decrease as can be seen in Figure 10b. That is because fewer training samples may lead to a less discriminative CNN architecture in distinguishing similar scenes.

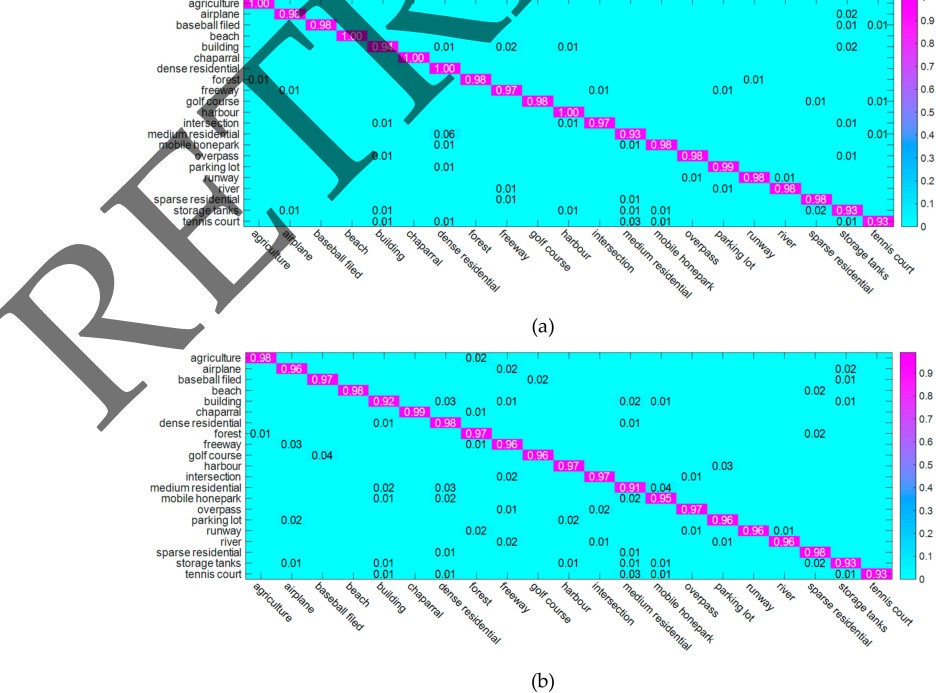

**Figure 10.** The confusion matrices of the ADFF approach for the UC Merced dataset; (**a**) 80% training ratio, (**b**) 50% training ratio.

### 4.3. Experimental Results on the AID Dataset

The ADFF framework outperforms other baseline algorithms with OAs of 94.75% and 93.68% by 50% and 20% training samples, respectively, according to Table 3. Conclusions similar to those in Table 2 can be drawn from Table 3. The classification accuracy of methods in the AID dataset is lower than that of the UC Merced dataset, since the AID dataset demonstrates the highest complexity and variations.

**Table 3.** Standard deviations and overall accuracy (%) of the ADFF framework and all baseline algorithms with 20% and 50% training ratio in the AID dataset.

| Methods | Published Year | 50% Training Ratio | 20% Training Ratio |
|---|---|---|---|
| The Proposed ADFF | 2019 | 94.75 ± 0.24 | 93.68 ± 0.29 |
| CaffeNet with DCF | 2018 | 93.10 ± 0.27 | 91.35 ± 0.23 |
| VGG-VD16 with DCF | 2018 | 93.65 ± 0.18 | 91.57 ± 0.10 |
| TEX-Net-LF | 2017 | 92.96 ± 0.18 | 90.87 ± 0.11 |
| CaffeNet | 2017 | 89.53 ± 0.31 | 86.46 ± 0.47 |
| GoogleNet | 2017 | 88.39 ± 0.55 | 85.44 ± 0.40 |
| VGG-16 | 2017 | 89.64 ± 0.36 | 86.59 ± 0.29 |
| Multi-Branch Neural Network | 2018 | 91.46 ± 0.44 | 89.38 ± 0.32 |
| Triplet Network | 2015 | 89.10 ± 0.30 | 86.89 ± 0.22 |
| BOCF | 2018 | 87.63 ± 0.41 | 85.24 ± 0.33 |
| salM3LBP-CLM | 2017 | 89.76 ± 0.45 | 86.92 ± 0.35 |
| Scene Capture | 2018 | 89.43 ± 0.33 | 87.25 ± 0.31 |
| Color-Boosted Saliency-Guided BOW | 2018 | 88.67 ± 0.39 | 86.67± 0.38 |

Different from the conclusions drawn in Table 2, the salM3LBP-CLM, scene capture and color-boosted saliency-guided BOW can provide classification accuracies comparable to several CNN architectures, including CaffeNet, GoogleNet, VGG-16 and triplet network. The reasons may be different for these three methods. For salM3LBP-CLM, the fusion of handcrafted features can distinguish those scene images that are easily confused in the case of insufficient data compared with CNN architectures. But for scene capture and color-boosted saliency-guided BOW, the label information and salient region information contained in feature representations can be beneficial to the scene classification of HRRSI.

As for the analysis of the confusion matrices shown in Figure 11b, 90% of all 30 categories achieve classification accuracies greater than 90% where the beach class achieves the highest accuracy 98.0%. Some categories that are similar in spectral characteristics, including "stadium", "sparse residential" and "port" are also classified accurately with 97.9%, 97.7% and 97.6%, respectively. The classes of "commercial" and "resort" had relatively low classification accuracies with 89.7% and 89.7%. In detail, the "commercial" class is easily confused with "dense residential" and "industrial" because they all constitute buildings, but their spatial distributions are different. In addition, the resort class is usually misclassified as "park" and "medium residential" due to the existence of some analogous objects, such as green belts and ponds. As can be seen in Figure 11a, with the increase of training samples, most categories increase their classification accuracies.

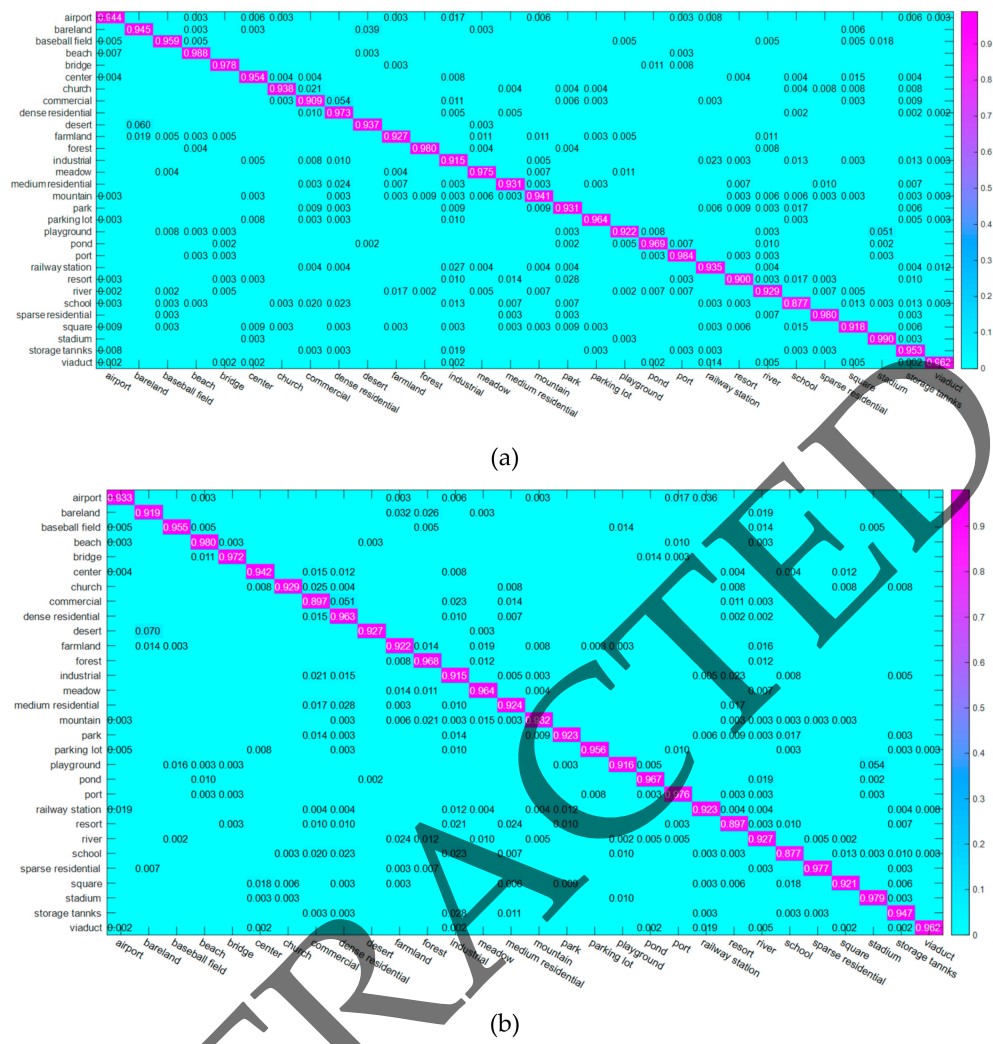

**Figure 11.** The confusion matrices of the proposed ADFF method in the AID dataset. (**a**) 50% training ratio. (**b**) 20% training ratio.

### 4.4. Experimental Results on the NWPU-RESISC45 Dataset

Table 4 shows the classification performance comparison using the most challenging NWPU-RESISC45 dataset. It can be observed that the ADFF method also achieves remarkable classification results in at least OA improvements of 1.35% and 1.99% in the case of 10% and 20% training samples. CaffeNet with DCF, VGG-VD16 with DCF, TEX-Net-LF and DenseNet can also obtain comparable classification accuracy compared with the proposed ADFF method, since they can force scenes with the same category to be close and those with diverse categories to be far away when training data is limited. The Hydra delivers a little higher classification accuracy than the proposed ADFF framework, since it fine-tunes a coarsely optimized ReseNet multiple times.

Different from the conclusions drawn in Tables 2 and 3, the CaffeNet, GoogleNet, VGG-16 and Multi-Branch Neural Network deliver classification accuracies lower than unsupervised feature learning-based methods in the NWPU-NESIS45 dataset. That is because it is not enough to train these CNN architectures for distinguishing similar scene images with only 20% training samples. The NWPU-RESISC45 dataset is with the lowest classification accuracy due to diverse scene categories and highly different spectral characteristics.

**Table 4.** Standard deviations and overall accuracy (%) of the ADFF framework and all baseline algorithms with 10% and 20% training ratio in the NWPU-RESISC45 dataset.

| Methods | Published Year | 20% Training Ratio | 10% Training Ratio |
|---|---|---|---|
| The Proposed ADFF | 2019 | 91.91± 0.23 | 90.58 ± 0.19 |
| CaffeNet with DCF | 2018 | 89.20± 0.27 | 87.59 ± 0.22 |
| VGG-VD16 with DCF | 2018 | 89.56 ± 0.25 | 87.14 ± 0.19 |
| TEX-Net-LF | 2017 | 88.37 ± 0.32 | 86.05 ± 0.24 |
| CaffeNet | 2017 | 79.85 ± 0.13 | 77.69 ± 0.21 |
| GoogleNet | 2017 | 78.48 ± 0.26 | 77.19 ± 0.38 |
| VGG-16 | 2017 | 79.79 ± 0.15 | 77.47 ± 0.18 |
| Multi-Branch Neural Network | 2018 | 76.38 ± 0.34 | 74.45 ± 0.26 |
| Triplet Network | 2015 | 88.01 ± 0.29 | 86.02 ± 0.25 |
| Hydra | 2019 | 94.51 ± 0.29 | 92.44 ± 0.34 |
| DenseNet | 2017 | 90.96 ± 0.31 | 89.38 ± 0.36 |
| BOCF | 2018 | 85.32 ± 0.17 | 83.65 ± 0.31 |
| salM3LBP-CLM | 2017 | 86.59 ± 0.28 | 85.32 ± 0.17 |
| Scene Capture | 2018 | 86.24 ± 0.36 | 84.84 ± 0.26 |
| Color-Boosted Saliency-Guided BOW | 2018 | 87.05 ± 0.29 | 85.16 ± 0.23 |

Figure 12 gives the confusion matrices generated from the classification results by the ADFF approach with 10% and 20% training samples. As shown in Figure 12a, 30 categories among all 45 categories achieve classification accuracies greater than 90%. The major confusion is in "rectangular farmland" and "circular farmland" because both of them have similar styles of farmlands, but with different shapes. With the decrease of training samples, the classification accuracy of almost all categories decreases correspondingly according to Figure 12b.

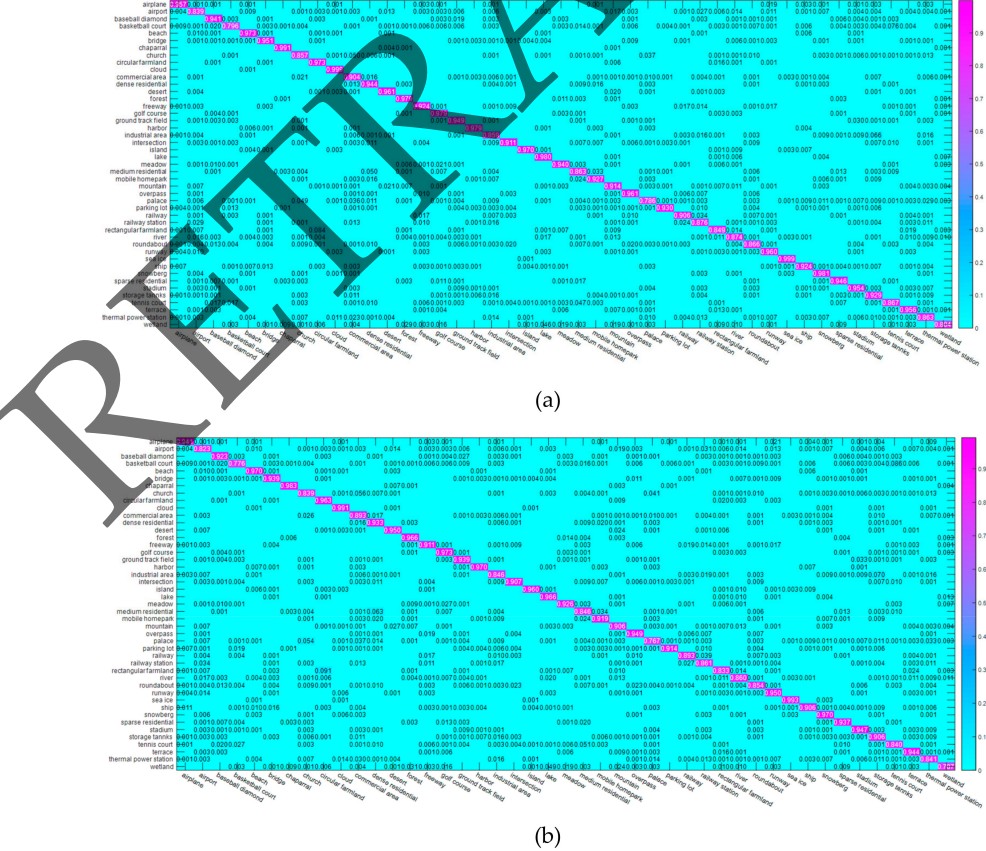

(a)

(b)

**Figure 12.** The confusion matrices of the proposed ADFF approach in the NWPU-RESISC45 dataset; (**a**) 20% training ratio, (**b**) 10% training ratio.

## 5. Discussion

### 5.1. The Computation Cost of the Proposed ADFF Framework

Table 5 shows the computation cost of the ADFF framework in three datasets under different ratios. When the number of training samples is 1050, 2000 and 5000 respectively, the computation cost will be 5349s, 10082s and 23274s.

**Table 5.** The computation cost of the ADFF approach in three datasets.

| The Training Ratio | UC MERCED | AID | NWPU-ESISC45 |
|---|---|---|---|
| 10% | - | - | 15,053 s |
| 20% | - | 10,082 s | 28,355 s |
| 50% | 5349 s | 23,274 s | - |
| 80% | 7983 s | - | - |

### 5.2. Ablation Studies of the Proposed ADFF Framework

Table 6 shows the classification accuracies of the proposed ADFF method, as well as its different parts. As can be seen in Table 6, attention maps, a multiplicative fusion of deep features and center loss are all effective in increasing classification accuracy with 7.63%, 5.85% and 3.79%. Attention maps play a more important role than a multiplicative fusion of deep features and center-based cross-entropy loss, since they can extract the key region that is significant for recognizing a scene. The multiplicative fusion of deep features is more significant than center loss because it can decrease the possibility of confusion in scenes of repeated texture.

**Table 6.** Standard deviations and overall accuracy of approaches removing one part of the ADFF framework with a higher training ratio.

| Methods | UC MERCED | AID | NWPU-ESISC45 |
|---|---|---|---|
| The Proposed ADFF | 97.53 ± 0.63 | 94.75 ± 0.24 | 90.91 ± 0.23 |
| Without Attention Maps | 90.46 ± 0.85 | 86.39 ± 0.45 | 83.44 ± 0.36 |
| Without Multiplicative Fusion of Deep Features | 91.63 ± 0.72 | 88.86 ± 0.40 | 85.16 ± 0.34 |
| Without the Center Loss, but only with Cross-Entropy Loss | 93.79 ± 0.68 | 90.87 ± 0.31 | 87.15 ± 0.30 |

As can also be seen in Table 6, the classification accuracy is still acceptable to three datasets if we only use the cross-entropy loss. That is because the cross-entropy loss may increase the discriminative ability of the CNN by pulling the probability distribution of predicted labels close to that of true labels. The center-based cross-entropy loss function performs better than the cross-entropy loss, since the center-based cross-entropy loss can better distinguish difficult samples while keeping the discriminative ability to distinguish samples that are easy to classify.

### 5.3. The Influence of Attention Maps on the Proposed ADFF Framework

Figure 13 shows the attention maps generated by the Grad-CAM algorithm. Note that the higher the brightness of the color is, the higher the importance of the corresponding area of the image is. As can be seen in Figure 13, the fine-tuned network learns well to exploit important information about recognizing one scene image, since the areas that the fine-tuned network focuses on are beneficial to the final output decision. Therefore, we can draw a conclusion that the attentions map derived from Grad-CAM have a positive influence on the scene classification information.

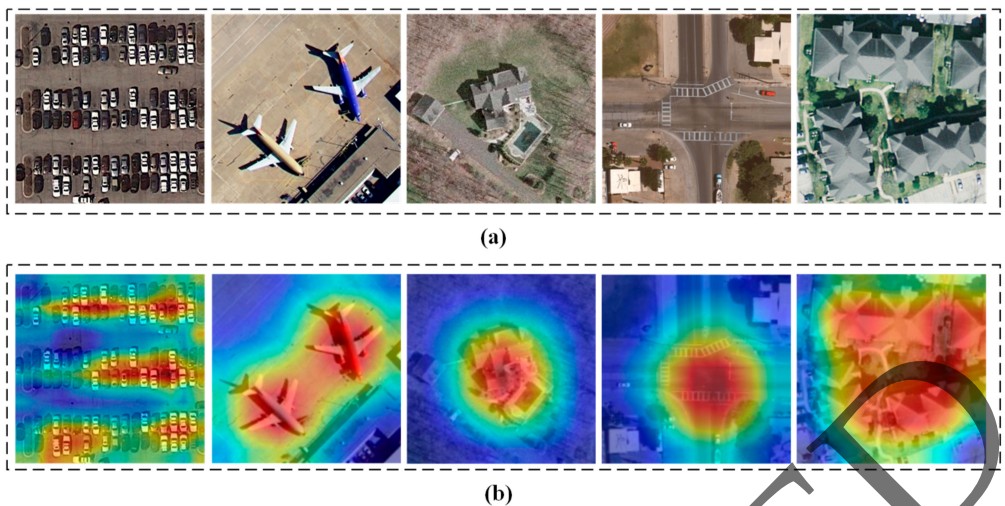

**Figure 13.** The attention maps of scene images derived from Grad-CAM. (**a**) The original scene image. (**b**) The Grad-CAM visualization results.

*5.4. The Limits of the Proposed ADFF Framework in Different Types of Images*

In order to analyze the limits of the ADFF framework in different types of images, the major confusion of the ADFF approach from three datasets is depicted respectively. As can be concluded from Figures 10 and 14, the confusion mainly exists in tennis court/medium residential, mobile home park/medium residential, golf course/baseball field, harbor/parking lot and building/dense residential in the UC Merced dataset. Different reasons cause the confusion of the proposed ADFF method. Scene images in Figure 14a,b,e are with different spatial distributions of the same objects. Figure 14c is with similar spectral characteristics in the background objects, such as grass. The vehicles in parking lot scenes are easily confused with ships in the harbor scenes, as shown in Figure 14d.

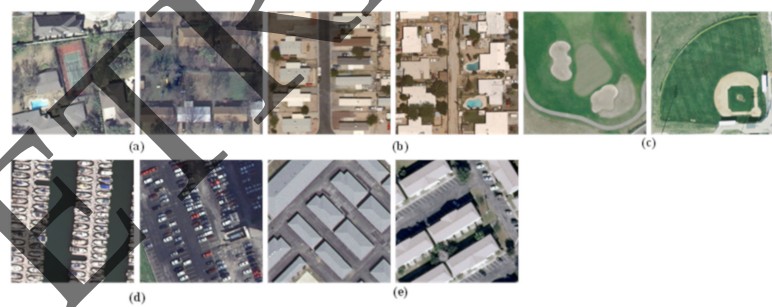

**Figure 14.** The major confusion of the proposed ADFF approach in the UC Merced dataset. (**a**) Tennis court and medium residential. (**b**) Mobile home park and medium residential. (**c**) Golf course and baseball field. (**d**) Harbor and parking lot. (**e**) Building and dense residential.

For the AID and NWPU-RESISC45 dataset, the major confusion of the ADFF framework is different from that in the UC Merced dataset as can be summarized from Figure 11 to Figures 12 and 15 to Figure 16. The major confusion exists in commercial/dense residential, desert/bare land, stadium/playground, airport/runway and bare land/farmland in the AID dataset. For the NWPU-RESISC45 dataset, the major confusion exists in basketball court/tennis court, circular farmland/rectangular farmland, church/commercial area, storage tanks/industrial area and roundabout/intersection.

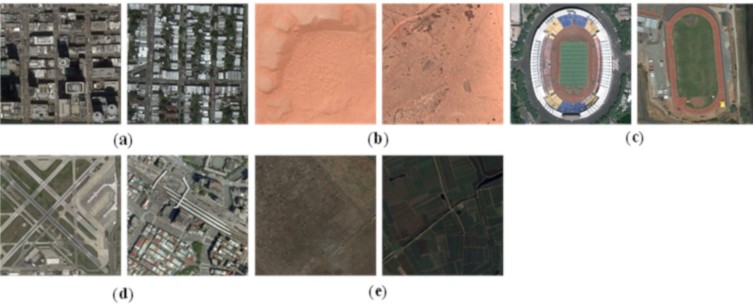

**Figure 15.** The major confusion of the ADFF approach in the AID dataset. (**a**) Commercial and dense residential. (**b**) Desert and bare land. (**c**) Stadium and playground. (**d**) Airport and railway station. (**e**) Bare land and farmland.

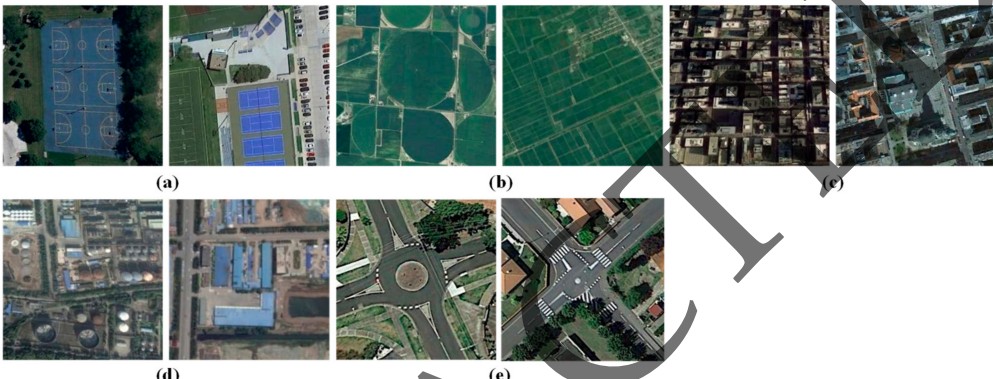

**Figure 16.** The major confusion of the ADFF approach in the NWPU-RESISC45 dataset. (**a**) Basketball court and tennis court. (**b**) Circular farmland and rectangular farmland. (**c**) Church and commercial area. (**d**) Storage tanks and industrial area. (**e**) Roundabout and intersection.

The reasons for the confusion are different: Figures 15a and 16c–d are all composed of buildings, but the spatial distribution of these buildings is different. Scenes of Figure 15b,c,e and Figure 16a–b share similar backgrounds. Figures 15d and 16e are with similar road networks in the scenes.

As can be concluded from Figure 14 to 16, the proposed ADFF framework is limited in handling the scenes that share close spectral characteristics, and those are with different spatial distributions of the same objects.

*5.5. The Limits of the Proposed ADFF Framework in Different Environments*

In order to analyze the limits of the proposed ADFF approach in different environments, we take the confusion of river scene images under different environments in Figure 17 as an example. As can be seen in Figure 17a,c,d,e,f, when the ground objects surrounding the river occupy a large proportion of the river scene images, these scene images are easily confused with the surrounded objects. As shown in Figure 17b, when the river is dry, it is easily confused with bare land that shares similar spectral characteristics.

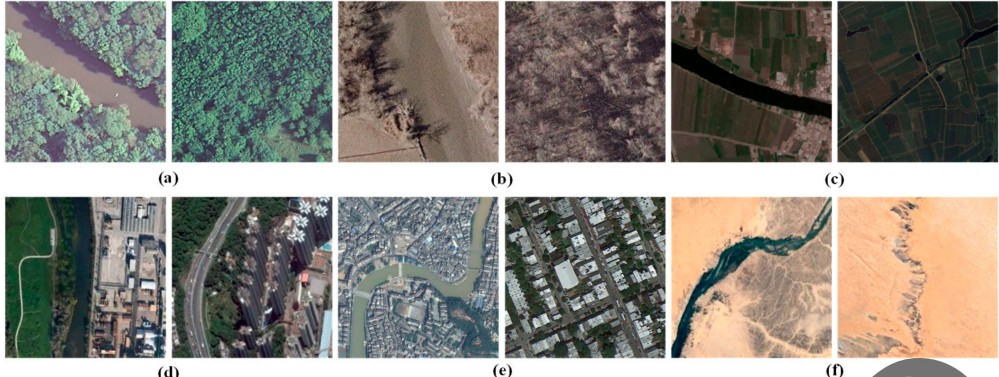

**Figure 17.** The major confusion of the ADFF approach in river under different environments; (**a**) river surrounded by forests, (**b**) dried rivers, (**c**) rivers surrounded by farmland, (**d**) rivers near the buildings and meadow, (**e**) rivers across the residential, (**f**) rivers in the desert.

As can be deduced from Figure 17, the ADFF framework may deliver limited performance in scene images where the objects surrounding the discriminative regions occupy a large area and those containing untypical salient objects, such as dry rivers or bald trees.

## 6. Conclusions

In this paper, an ADFF method is proposed to reduce the influence of intra-class variations and repeated texture on the scene classification. In this method, attention maps derived from Grad-CAM approach serve as an explicit input in order to make the network focus on salient regions beneficial to scene classification. Then deep features from the attention maps and original RGB images are fused by multiplicative fusion for better performance in scenes of repeated texture. Finally, the center-based cross-entropy loss is proposed to reduce the confusion in scene images difficult to classify.

The proposed ADFF framework is evaluated on three large three benchmark datasets to prove its effectiveness in scene classification. Several conclusions can be drawn from the experiments.

First of all, the classification accuracy of the ADFF approach outperforms other competitive scene classification methods with an overall accuracy of about 97% when the training ratio is large.

Secondly, the ADFF approach can still achieve a competitive accuracy of 91% in the case of limited training data. Therefore, it can be applied to the land-cover classification in a large area in the case of limited training data.

Last, but not least, attention maps, multiplicative fusion of deep features, and the center-based cross-entropy loss function are also proved to be effective in increasing an average classification accuracy of 3.3%, 5.1%, and 6.1%, respectively.

Nevertheless, the proposed ADFF approach demonstrates its limitation in providing the boundary information of the land-cover types. Therefore, the fusion of a scene-level land-cover classification method with a pixel-level or object-based land-cover classification method needs to be investigated in the future.

**Author Contributions:** Conceptualization, R.Z.; methodology, R.Z.; writing—original draft preparation, R.Z. and N.M.; writing—review and editing, R.Z., N.M. and Y.L.; supervision, Y.L. and L.Y.; funding acquisition, L.Y.

**Funding:** This research is funded by The National Key Research and Development Program of China under grant no. 2017YFC0803802.

**Conflicts of Interest:** The authors declare no conflict of interest.

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
