# Peer review of "Attention-Based Deep Feature Fusion for the Scene Classification of High-Resolution Remote Sensing Images"

_remotesensing, doi:10.3390/rs11171996_

Round 1
Reviewer 1 Report
** The English is more or less good. However, it needs still some editing. There are still some grammatical issues (not hampering the understanding though).
** In Section 3.2 while discussing HSSI images please refer Section 4 (where you discuss the exact datasets). I was not sure what datasets you have used while I was at Section 3.2.
** One major issue I have is the fact that you have used Grad-CAM as it is. However, attention depends on the use-case or problem being considered. Attention also depends on the data we are using. One of the major contributions of the paper can be the discussion on how to finetune Grad-CAM algorithm. The Grad-CAM algorithm is definitely not use-agnostic or data-agnostic. Remote-sensing is very different from conventional image-processing. Hence, finetune of Grad-CAM need to be discussed in detail.
** Before Equation (4) it has been mentioned that "cross-entropy loss function" is frequently applied. This needs to be bolstered with some citations. Secondly, frequent use does not mean that it is the best tool for the job. A more theoretically sound discussion of the effectiveness of cross-entropy is strongly suggested.
** In abstract (as well as in the main paper), I would suggest not to use the phrase "superiority". By no-free-lunch rule, no algorithm is exactly superior. E.g. in the last two lines of abstract it can be mentioned that "...benchmark datasets to show the performance..".
** A table summarizing all the datasets in Section 4 would be nice.
** In addition to the tests, it will be really pertinent to run your algorithm on a bigger region (e.g. a district in China) and show that it actually works in real-life situations. As remote-sensing engineers, we are more interested in how good your algorithm is in real-life situations.
** Lastly, also compare the results with your recent work "Class-Specific Anchor Based and Context-Guided Multi-Class Object Detection in High Resolution Remote Sensing Imagery with a Convolutional Neural Network".
Author Response
Dear Editors and reviewers,
We are particularly grateful to you and the anonymous reviewers for the careful reading and constructive comments.
According to the comments, we have tried our best to revise the manuscript to make it better, and an item-by-item response follows in the attachment. The modified parts have been highlighted in yellow color in the revised manuscript. Thanks very much for your time.
Best regards
Ruixi Zhu

Reviewer 2 Report
The work presented in this paper is of interest to scientists working on the classification of radar images. the overall organization of the paper is correct. however, it would be helpful to consider the remarks below.
- the introduction should be reworked to highlight the contribution of the paper, by further specifying the motivation, the problem and the methodology adopted, and it's necessary to mention the quantified objectives to be achieved.
- it would be useful to include also recent references on the subject. - it would be useful to specify the calculation time required for the execution of the adopted methodology, at least at the level of the simulations carried out.
- it would be useful to challenge the limits of the adopted methodology, according to the type of images, the environment, the speckle, ...
- it would be useful to specify the implementation of the methodology in a practical context and / or operational.
- the conclusion should be reinforced by adding the perspectives offered to the work done and presented in this paper.
Author Response

(The authors gave the same response as above.)

Reviewer 3 Report
Figure 4 and the confusion matrices of the ADFF (figs. 10-12) replaced by a vector graphic instead of used bitmaps in order to improve their readability. This can be also done in most of the figures in the paper where labels are appeared. Add also the computational cost of your system. Please, provide a webpage (in the paper) that will include a link to the used dataset and your results for people that are interesting for comparisons. This work should be compared with state of the art works [2] and [5] at least under NWPU-RESISC45 dataset. In [2] you can find the reported results on this datasets. The reasons of publication of this work (contribution) should be better explained. If it also possible, please apply your method on public datasets for building detection , you can use SZTAKI-INRIA building detection dataset (see Benedek et al. (2012)). http://web.eee.sztaki.hu/remotesensing/building_benchmark.html In addition, in order to improve your related work, you can cite the following related works.[1] Zhu, X. X., Tuia, D., Mou, L., Xia, G. S., Zhang, L., Xu, F., & Fraundorfer, F. (2017). Deep learning in remote sensing: A comprehensive review and list of resources. IEEE Geoscience and Remote Sensing Magazine, 5(4), 8-36.
[2] Minetto, Rodrigo, Maurício Pamplona Segundo, and Sudeep Sarkar. "Hydra: an ensemble of convolutional neural networks for geospatial land classification." IEEE Transactions on Geoscience and Remote Sensing (2019).
[3] Benedek, C., Descombes, X., Zerubia, J., 2012. Building development monitoring in multitemporal remotely sensed image pairs with stochastic birth-death dynamics. IEEE Trans. Pattern Anal. Mach. Intell. 34, 33–50.
[4] I. Grinias , C. Panagiotakis and G. Tziritas, MRF-based Segmentation and Unsupervised Classification for Building and Road Detection in Peri-urban Areas of High-resolution, ISPRS Journal of Photogrammetry and Remote Sensing, vol. 122, pp. 145-166, 2016.
[5] G. Huang, Z. Liu, L. van der Maaten, and K. Q. Weinberger, “Densely connected convolutional networks,” in IEEE Conference on Computer Vision and Pattern Recognition (CVPR), 2017, pp. 4700– 4708
The English of the paper should be improved. The paper should be corrected by a native speaker.
Author Response
Dear Editors and reviewers,
We are particularly grateful to you and the anonymous reviewers for the careful reading and constructive comments.
According to the comments, we have tried our best to revise the manuscript to make it better, and an item-by-item response follows in the attcahment. The modified parts have been highlighted in yellow color in the revised manuscript. Thanks very much for your time.
Best regards
Ruixi Zhu

Round 2
Reviewer 1 Report
Thanks for the prompt response and modifications. I am happy with the current version.
Author Response
Thank you for your suggestions and your effort on reviewing my article.
Reviewer 2 Report
The quality of the images is to be improved and to avoid certain errors of syntax and style it would be necessary to reread completely the contents of the article
Author Response
First of all, thank you for your suggestions and effort on reviewing my article.
We have improved the quality of images by raplacing the ".tif" with ".emf" and modifying the fontsize in the images to make image clearer.
We have also reread completely the contents of the article carefully in order to rule out possible errors of syntax and style.
This manuscript is a resubmission of an earlier submission. The following is a list of the peer review reports and author responses from that submission.